# Interdental Papillary Reconstruction by Microtunnelling Technique Using Autologous Biomatrices—A Randomised Controlled Clinical Trial

**DOI:** 10.3390/medicina58101326

**Published:** 2022-09-22

**Authors:** Sindhura Gadi, Sangeetha Subramanian, P. S. G. Prakash, Devapriya Appukuttan, Abirami Thanigaimalai, Maha A. Bahammam, Khalid J. Alzahrani, Khalaf F. Alsharif, Ibrahim F. Halawani, Mrim M. Alnfiai, Thodur Madapusi Balaji, Shankargouda Patil

**Affiliations:** 1Department of Periodontics, S.R.M. Dental College & Hospital, Ramapuram, Chennai 600089, Tamil Nadu, India; 2Department of Periodontics, Priyadarshini Dental College and Hospital, Chennai 600029, Tamil Nadu, India; 3Department of Periodontology, Faculty of Dentistry, King Abdulaziz University, Jeddah 21589, Saudi Arabia; 4Executive Presidency of Academic Affairs, Saudi Commission for Health Specialties, Riyadh 11614, Saudi Arabia; 5Department of Clinical Laboratories Sciences, College of Applied Medical Sciences, Taif University, P.O. Box 11099, Taif 21944, Saudi Arabia; 6Department of Information Technology, College of Computers and Information Technology, Taif University, P.O. Box 11099, Taif 21944, Saudi Arabia; 7Department of Dentistry, Tagore Dental College and Hospital, Chennai 600127, Tamil Nadu, India; 8College of Dental Medicine, Roseman University of Health Sciences, South Jordan, UT 84095, USA

**Keywords:** black triangle, connective tissue graft, interdental papillary loss, papilla reconstruction, platelet-rich fibrin, surgical microscope

## Abstract

*Background and objectives*: The study aimed to evaluate and compare the amount of papillary gain and black triangle height reduction after intervention with a microtunnelling technique with either Connective tissue graft (CTG) or Platelet-rich fibrin (PRF) as a biomatrix at 6 months using a microsurgical approach. *Materials and Methods:* Twenty-six patients with interdental papillary loss were included in the study. The patients were selected randomly for the study groups with thirteen patients in each group: a control group where CTG was utilised as a matrix, and a test group where PRF was utilised as a matrix, for interdental papillary reconstruction. A microtunnelling technique was performed for both the study groups under a surgical microscope. The primary parameters assessed were interdental Papillary height (PH) and Black triangle height (BTH) at baseline, with secondary parameters Visual analogue score by dentist (VAS-D) and patient (VAS-P) assessed at 6 months. *Results:* Both the control and test groups showed a significant reduction in BTH within their respective group at six months (*p* < 0.05). The gain in papillary height significantly improved only in the CTG group at 6 months. However, significant differences could not be demonstrated for any of the variables such as BTH (*p* value = 0.582) and PH (*p*-value = 0.892) between the study groups at 6 months. *Conclusions:* IDP reconstruction utilising a microtunnelling approach with CTG or PRF was successful without any significant differences between the groups for the parameters assessed at 6 months.

## 1. Introduction

An attractive smile is an essential feature that influences an individual’s personality and social interaction. Increasing aesthetic demands have brought about a requirement to maintain and restore the different parts of the gingival complex, especially the interdental papilla. Unfortunately, the loss or absence of interdental papilla (IDP) between the two adjacent teeth results in unaesthetic black triangles. Black triangles are found to be less attractive and this remains a concern for both dentists and patients. Factors affecting the interdental papillary fill are traumatic oral hygiene practices [1], papillary loss associated with plaque [2], improper contours of restoration, loss of teeth/spacing between the teeth, underlying osseous support [3], gingival biotype and bioform [4], and the anatomy of the tooth and its contact points [5,6,7,8]. Surgical management of deficient interdental papilla includes various techniques in the literature, which are based on flap elevation/reflection procedures [9,10]. Flap elevation has the downside of papillary tear and it also compromises the vascularity. The results obtained were not predictable and lacked stability over a long-term period. To overcome this, tunnelling-based augmentation procedures have been proposed, which do not require the papilla to be incised and, hence, do not compromise the blood supply to a great extent.

Some studies have incorporated biomaterials along with the papillary augmentation technique to counter the dead space, which further enhances the healing and regenerative potential of the papilla. CTG is regarded as the gold standard biomatrix for the augmentation of periodontal soft tissue procedures, including the papillary augmentation procedure [11,12,13]. However, CTG also comes with several disadvantages such as a second surgical site to harvest the graft, patient discomfort, and increased surgical time. 

This has led to the use of alternate autogenous matrices that are comparable in their effects to overcome the disadvantages of CTG. One such biomaterial is Platelet-rich fibrin (PRF), which has reported adequate papillary fill when utilised for IDP reconstruction [14,15,16].

Since case reports and series are in favour of these two biomatrices, namely, CTG and PRF for augmentation of IDP, they were considered for the present randomised controlled clinical trial. CTG, the gold standard matrix was allocated to the control group and PRF to the test group. Conventional IDP reconstruction procedures are technique sensitive because of limited working space and restricted blood supply to the area. Magnification of the IDP with the help of a surgical microscope will increase the visibility and permit gentle handling of the tissues with microsurgical instruments. Microsurgical procedures also prevent inadvertent severity of the soft tissues and decrease postoperative discomfort as well as enhance wound healing and increase patient satisfaction. These factors can enhance collectively the regenerative potential of the IDP. Hence, we have utilised the advantage of the surgical microscope for this study.

The current microsurgical approach has been designed as a randomised controlled clinical trial (RCT) with the aim to compare CTG and PRF in a microtunnelling technique to augment papillary dimensions. The hypothesis of the study is that PRF will augment papillary dimensions favourably comparable to that of CTG in the proposed technique when the surgery is performed under a surgical microscope.

## 2. Materials and Methods

The study has been designed as a randomised controlled clinical trial to assess the regenerative potential of autologous biomatrices in the microtunnelling technique for interdental papillary reconstruction. Sample size calculation was performed fixing the alpha error at 5% with 80% power [17,18], and the required sample was estimated to be a total of 26 patients with 13 patients in each of the two groups. Expecting a dropout of 20% or more, 32 patients were initially enrolled for the study at the onset. The microtunnelling technique for papilla reconstruction was proposed for both groups. CTG and PRF were used as the biomatrices for the control group and the test group, respectively. The study was conducted according to the guidelines of the Declaration of Helsinki and the study protocol was approved by the Institutional Scientific and Ethical Review Board. The ethical approval number of the current study is SRMDC/IRB/2018/MDS/NO.505. Clinical Trial Registry India received registration for the study under reference number CTRI/2019/09/021128.

Patients satisfying the following criteria were recruited: systemically healthy patients with Nordland and Tarnow’s Class I, II, and I–II interdental papillary loss, gingival recession < 2 mm on the facial aspect, patients devoid of periodontal disease, patients presenting with adequate width of attached gingiva. Patients excluded were: smokers, pregnant/lactating women, patients with a history of periodontal surgery within the last six months and with platelet disorders, patients with insufficient plaque control, and those not willing to undergo surgery.

Patients who exhibited full mouth plaque scores and bleeding scores ≤20% after nonsurgical periodontal therapy were considered for surgery. Patients were randomised into control or test groups according to the last digit of their outpatient (OP) card number. The number was written on a paper and placed in a sealed envelope and presented to the surgeon at the time of surgery. For odd numbers, CTG was utilized, and for even numbers, PRF was utilized as the biomatrix.

The clinical parameters evaluated were plaque scores (O’Leary plaque index), bleeding scores (Ainamo and Bay), Probing pocket depth (PPD), site-specific PPD and Clinical attachment level (CAL), site-specific CAL, height of the black triangle, and interdental papillary height. The measurement from the tip of the interdental papilla to the apical-most point of the contact point was taken as the height of the black triangle, and the measurement from the base of the papilla (a line drawn from the zenith of the adjacent gingival margin) to the tip of the interdental papilla was considered as the interdental papillary height. All the clinical parameters were assessed at baseline and at six months by an experienced examiner who was masked to the study (Figure 1).

Recruitment of the patients was undertaken after assessing the mean PPD, CAL, and interpapillary loss [19]. All the other clinical measurements were assessed at both baseline and six months. At the conclusion of six months, VAS ratings were marked on a scale of 1 to 10 by patients. Similarly, dentists also assessed VAS scores based on papillary fill. Each patient’s preoperative and postoperative photos were positioned with an equal aspect ratio. Based on the improvement in the papillary fill, the patients gave their scores as indicated by the given scale. Similarly, the dentist was asked to rate the papillary fill on a scale of zero to ten with zero being least satisfied and ten being most satisfied. Two periodontists who were blinded to the surgery scored the VAS independently by evaluating the baseline and postoperative photographs of each patient. All the surgical procedures were carried out by a single skilled operator.

### 2.1. Surgical Procedure

After administering local anaesthesia, the surgical procedure was performed under a surgical microscope with 10× magnification. A 3 mm horizontal incision was made apical to the base of the interdental papilla adjacent to the mucogingival junction. Without splitting the interdental papilla, a sulcular incision was made along the buccal and palatal region of the neighbouring teeth. Using tunnelling knives, the buccal aspect of the involved interdental gingiva was tunnelled from the horizontal incision and continued over the crest and extended to the palatal aspect. The biomatrix CTG was utilised for the control group and PRF was utilised for the test group. A single incision approach was used to harvest the connective tissue graft from the palate. PRF was procured by centrifuging 10 mL of intravenous blood at 3000 rpm for 13 min, which was then flattened to form a membrane. The matrices thus obtained were placed on the buccal side through the horizontal incision and passed over the crest of the interdental bone so that it covered both the buccal and palatal aspects of the interdental bone. Light-cured flowable composite resin was placed in the coronal point of the contact area and cured. Vertical mattress suture was given on the interdental papilla, which was suspended on the composite resin (Figure 2 and Figure 3). Periodontal dressing was used to cover the surgical site. Patients were instructed to take anti-inflammatory medications for 3 days. The patients were reviewed 2 weeks after the surgery for removal of sutures.

### 2.2. Statistical Analysis

SPSS (IBM SPSS Statistics for Windows, Version 26.0, Armonk, NY: IBM Corp. Released 2019) was used to perform data analysis. For intragroup comparison of the values of FMBS and FMPS, paired *t*-test was applied, whereas for intergroup comparisons, an independent sample *t*-test was applied. Chi-Square test was employed to compare proportions between the test and control groups; if any expected cell frequency was below five, Fisher’s exact test was used. Wilcoxon Signed Rank test was performed to compare values between baseline and 6 months. Mann–Whitney U test was used to compare values between the groups for the variables that did not follow a normal distribution.

## 3. Results

The clinical parameters of the CTG and PRF groups are represented in Table 1 as descriptive statistics. For parameters at the surgical site, such as PH, BTH, CAL, and PPD, the mean and standard deviation was given. The other parameters FMBS and FMPS were indicated as percentages. The findings revealed no significant variations in the parameters between the two groups at the time of baseline evaluation. 

In the intragroup comparison of site-specific PPD (mm) and CAL (mm), BTH showed significant differences in both the study groups at six months, except for PH, which was not significant in the PRF group. However, an intergroup comparison of all these clinical parameters did not show a significant difference when evaluated at 6 months (Table 2). 

The primary as well as secondary outcome variables of the study were compared between the groups at six months using the Mann–Whitney U Test (Table 3). No significant difference was found in the primary outcomes, such as PH and BTH, and the secondary outcomes, such as VASD and VASP, between the study groups.

A comparison of interpapillary height gain and decrease in black triangle height between the CTG and PRF groups at 6 months is shown in Table 4. A Mann–Whitney U Test was performed to compare the all the variables. No significant difference with regard to gain in PH and decrease in BTH was demonstrated in the study groups.

## 4. Discussion

The interdental papilla is a critical region with restricted space, limited access and blood supply. Hence, it has become a highly challenging task for the dentist to attain a predictable outcome with reconstruction techniques. Numerous nonsurgical and surgical techniques have been proposed in the literature with varied outcomes. Predictability of the postsurgical outcome is favourable when biomaterials are used along with IDP reconstruction techniques.

The present study utilised a microtunnelling technique to preserve as much vascularity as possible so that regeneration would be enhanced. The technique also facilitated the advancement of the entire interdental papilla as a gingivopapillary unit. Nevertheless, IDP is usually very delicate, and gentle manipulation of the tissues is very important to preserve the papilla. Hence, surgical magnification and microsurgical instruments were used to increase visibility and prevent inadvertent severing of the delicate interdental papilla. In this context, the current clinical trial was designed to compare CTG and PRF as biomatrices in the augmentation of a deficient interdental papilla in the maxillary anterior region using a microsurgical approach.

The recruited patients were evaluated for PH and BTH at baseline, which were the primary outcome measures of the study, and no significant difference was found when these parameters were compared between the study groups (Table 1).

Post surgically, again PH and BTH were evaluated at 6 months in both groups to determine the effectiveness of the biomaterials. In the CTG group, significant improvement was observed in the primary parameters at 6 months (Figure 2). This was not the case for the PRF group, where PH showed improvement from baseline but this, however, was not significant at 6 months (Figure 3).

When BTH was considered, both groups showed significant closure of the black triangle at 6 months from baseline. In addition, CTG was more significant in achieving the obliteration of the black triangle than PRF (Table 2). Additionally, our prior research also showed that at six months, the CTG group experienced a considerably greater reduction in BTH [20].

The findings of the present study once again emphasise the fact that CTG is the gold standard matrix for the augmentation of IDP. There is inherent regenerative potential in CTG, which improves the relevant parameters of the reconstruction technique. On the other hand, though PRF also substantially contributes to the augmentation of the papilla, it does not show a significant gain in PH at 6 months. The biological characteristics of PRF are such that they lack rigidity and, hence, may tend to collapse easily at the surgical site. This may in turn lead to a compromise in the space-making effect of PRF. This factor seems to be a significant concern for the regeneration of soft delicate tissues. Therefore, the efficacy of PRF in the proposed technique is slightly lower in attaining PH, and not comparable to CTG. However, there was no significant difference at 6 months between the CTG and PRF groups when the gains in PH and reduction in BTH were evaluated (Table 4).

Both the control and the test groups had substantial improvement from baseline to six months when the site-specific PPD and CAL were examined. However, when these parameters were compared between the study groups, no significant difference was found at 6 months. Visual analogue score recorded by dentist (VAS-D) and Visual analogue score recorded by patient (VAS-P) between the CTG group and PRF groups were more in favour of the CTG group. However, significant differences between the groups were not established as indicated by statistical analysis. 

All the primary as well as secondary clinical outcomes between the CTG and PRF groups at 6 months showed no significant difference, indicating that both CTG and PRF are comparable even though the values of each parameter were slightly favourable towards the CTG group at 6 months. This is in accordance with Singh et al.’s study, which employed a split thickness flap for IDP reconstruction, and at 3 months, significant variations in the parameters could not be demonstrated between the CTG and PRF groups.

The small sample size and follow-up duration of only 6 months are the shortcomings of the present study. Additionally, an evaluation of the distance between the radiographic bone crest to the contact point was not taken into account for the selection criteria. Within the limitations of the present microsurgical study, the results demonstrated that both CTG and PRF are effective in gaining papillary height and reducing black triangle height with the microtunnelling technique at the end of 6 months.

## 5. Conclusions

The results of the clinical trial concluded that the microtunnelling technique performed using a surgical microscope revealed slightly favourable outcomes with CTG; although, significant variations could not be demonstrated for papillary height gain and black triangle height reduction between the CTG and PRF groups at 6 months.

## Figures and Tables

**Figure 1 medicina-58-01326-f001:**
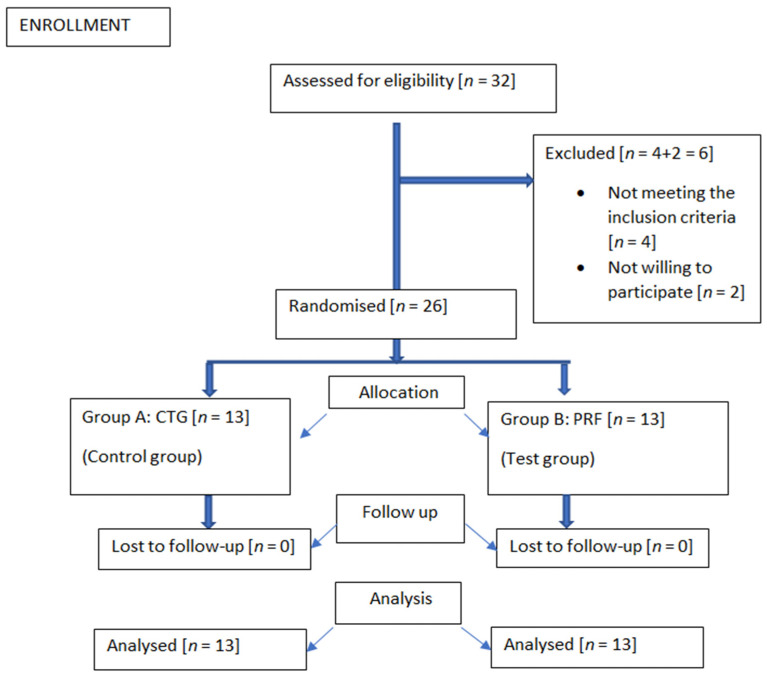
Consort Diagram.

**Figure 2 medicina-58-01326-f002:**
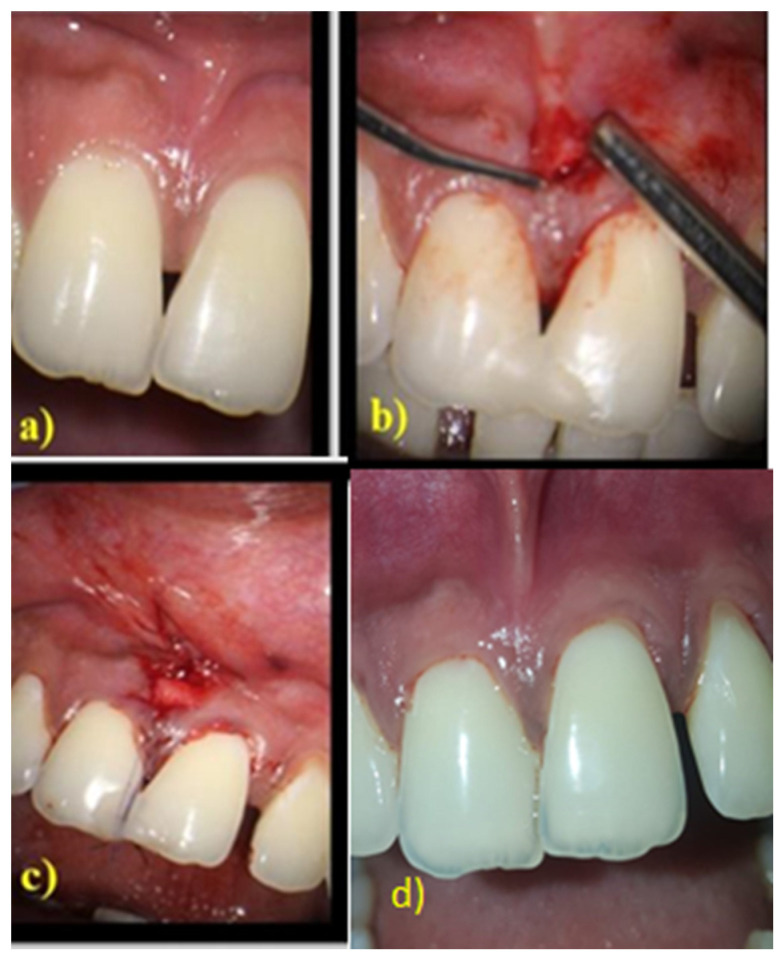
Surgical protocol of control group: (**a**) Preoperative image of 11 and 21 in CTG group. (**b**) Tunnel preparation in CTG group. (**c**) Vertical mattress sutures placed along the composite stopper in the CTG group. (**d**) Six months postoperative image in the CTG group. CTG: Connective Tissue Graft.

**Figure 3 medicina-58-01326-f003:**
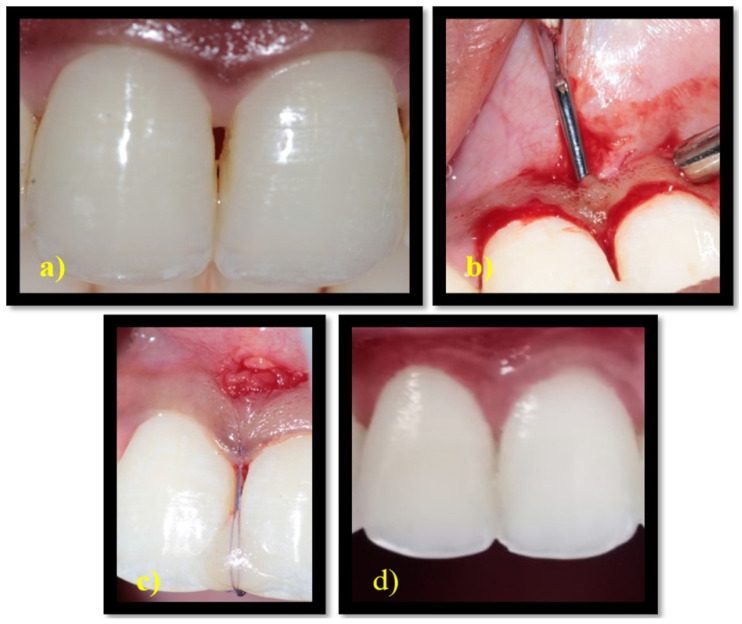
Surgical protocol of test group: (**a**) Preoperative image of 11 and 21 in PRF group. (**b**) Tunnel preparation in PRF group. (**c**) Vertical mattress sutures placed along the composite stopper in the PRF group. (**d**) Six months postoperative image in the PRF group. PRF: Platelet Rich Fibrin.

**Table 1 medicina-58-01326-t001:** Descriptive statistics of clinical parameters between the study groups.

Parameters/Groups	CTG Group (Mean ± SD)	PRF Group (Mean ± SD)	*p*-Value
Age	36 + 8.9	40.23 + 14.7	0.388
PPD (mm)	2.28 + 1.10	2.24 + 0.94	0.959
CAL (mm)	2.32 + 1.31	2.72 + 1.38	0.486
FMPS (%)	12.57 + 3.02	14.03 + 2.41	0.186
FMBS (%)	11.91 + 1.96	12.01 + 3.31	0.926
PH	3.42 + 1.51	3.62 + 0.94	0.359
BTH	3.00 + 0.54	2.58 + 0.79	0.077

CTG Connective Tissue Graft; PRF:Platelet Rich Fibrin; PPD: Probing Pocket Depth; CAL:Clinical Attachment Level; FMPS:Full Mouth Plaque Scores;FMBS: Full Mouth Bleeding Scores; PH:Papillary Height; BTH:Black Triangle Height.

**Table 2 medicina-58-01326-t002:** Intragroup and intergroup assessment of clinical parameters of the study groups.

Parameters	PRF Group	CTG Group	*p*-Value
PPD Baseline	2.24 + 0.94	2.28 + 1.10	0.959
PPD 6 months	1.87 + 0.64	1.75 + 0.55	0.676
*p*-Value	0.027	0.045	
CAL baseline	2.72 + 1.38	2.32 + 1.31	0.486
CAL 6 months	2.25 + 1.05	1.65 + 0.99	0.177
*p*-Value	0.018	0.023	
PH baseline	3.62 + 0.94	3.42 + 1.51	0.359
PH 6 months	4.19 + 0.83	4.23 + 0.95	0.892

**Table 3 medicina-58-01326-t003:** Sixth-month evaluation of primary and secondary outcome variables between the two groups.

Parameters/Groups	CTG Group (Mean ± SD)	PRF Group (Mean ± SD)	*p*-Value
Papillary height (mm)	4.23 + 0.95	4.19 + 0.83	0.892
Black triangle height (mm)	1.73 + 1.13	2.04 + 1.23	0.582
Visual analogue score—dentist	7.45 + 1.13	6.77 + 1.51	0.156
Visual analogue score—patient	7.15 + 1.21	6.08 + 1.98	0.128

*p* ≤ 0.05 statistically significant.

**Table 4 medicina-58-01326-t004:** Comparison of gain in papillary height and black triangle height reduction between the groups at 6 months.

Parameters/Groups	CTG Group (Mean ± SD)	PRF Group (Mean ± SD)	*p*-Value (*p* ≤ 0.05)
Gain in papillary height (mm)	0.85 + 0.88	0.77 + 0.56	0.937
Decrease in black triangle height (mm)	1.31 + 1.09	0.62 + 0.65	0.107

*p* ≤ 0.05 statistically significant.

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
