# Peer review of "Interdental Papillary Reconstruction by Microtunnelling Technique Using Autologous Biomatrices—A Randomised Controlled Clinical Trial"

_medicina, 2022, doi:10.3390/medicina58101326_

Round 1

Reviewer 1 Report

Dear authors,

thanks for having provided such an interesting manuscript.

It deals with the comparison of two different autologous biomatrices for papillary reconstruction.

Mine are simply minor observations as the manuscript is appropriately conducted from a methodological point of you.

1. Do not report results in materials and methods (e.g. line 99-100 must be moved in results).

2. English needs to be strongly revised as many typos and errors are present.

3. Be sure that every acronym is reported "in extenso" the first time you use it. (e.g. CTG in line 72)

4. Figure 1 has a very bad graphical release that prevents the reader from completely understanding it. Please fix it.

Regards

Author Response

Corrections as suggested by the reviewers

Reviewers

Comments

Corrections

Comment M1

Similar index in Red should be less than 5% and the total similar index should be lower than 35%

Similar index in Red has been corrected

Comment M2

“p”  has been corrected to italic font style “p” throughout the manuscript

Comment M3

In the image, please add a space before and after “=”. Also, revise “n” into the “n”.

Space added  before and after “=”.

“n” changed into the “n”.

1

Do not report results in materials and methods (e.g. line 99-100 must be moved in results).

It was ensured that results are not reported in materials and methods. Line 99-100  corresponds to the introduction section in the PDF document

1

English needs to be strongly revised as many typos and errors are present

English revision done

1

Be sure that every acronym is reported "in extenso" the first time you use it.

Acronym is expanded when it is used for the first time.

1

Figure 1 has a very bad graphical release that prevents the reader from completely understanding it. Please fix it.

Figure 1 replaced

2

There is no hypothesis

Hypothesis given in the last line of the introduction

2

 The introduction should be more focused on the topic of the study

Introduction refined  and focused on the topic

2

Smaller sample size and a follow-up duration of only 6 months are the shortcomings of the present study

This has been included in the last paragraph of the  discussion as a limitation

2

Poor quality of the images.

Figure 2d replaced

Reviewer 2 Report

 The  topic is interesting .

The study has been designed as a randomised controlled clinical trial to assess  the regenerative potential of autologous biomatrices in the microtunneling technique for  interdental papillary reconstruction

There is no hypothesis

  The introduction should be more focused on the topic of the study

Smaller sample size and a follow-up duration of only 6 months are the shortcomings of the present study

Poor and very poor quality of the images.They should be much more representative

Typographical errors that give the impression that the work has been very rushed

Author Response

(The authors gave the same response as above.)

Round 2

Reviewer 2 Report

the changes have been made properly